# Clinical risk calculators informing the decision to admit: A methodologic evaluation and assessment of applicability

**Neeloofar Soleimanpour**[1], **Maralyssa Bann**[2,3]*

1 Colorado School of Public Health, Aurora, Colorado, United States of America, 2 Department of Medicine, University of Washington School of Medicine, Seattle, Washington, United States of America, 3 Department of Medicine, Harborview Medical Center, Seattle, Washington, United States of America

* mbann@uw.edu

**Data Availability Statement:** All relevant data are within the manuscript and its Supporting information files.

**Funding:** The authors received no specific funding for this work.

## Abstract

### Introduction

Clinical prediction and decision tools that generate outcome-based risk stratification and/or intervention recommendations are prevalent. Appropriate use and validity of these tools, especially those that inform complex clinical decisions, remains unclear. The objective of this study was to assess the methodologic quality and applicability of clinical risk scoring tools used to guide hospitalization decision-making.

### Methods

In February 2021, a comprehensive search was performed of a clinical calculator online database (mdcalc.com) that is publicly available and well-known to clinicians. The primary reference for any calculator tool informing outpatient versus inpatient disposition was considered for inclusion. Studies were restricted to the adult, acute care population. Those focused on obstetrics/gynecology or critical care admission were excluded. The Wasson-Laupacis framework of methodologic standards for clinical prediction rules was applied to each study.

### Results

A total of 22 calculators provided hospital admission recommendations for 9 discrete medical conditions using adverse events (14/22), mortality (6/22), or confirmatory diagnosis (2/22) as outcomes of interest. The most commonly met methodologic standards included mathematical technique description (22/22) and clinical sensibility (22/22) and least commonly met included reproducibility of the rule (1/22) and measurement of effect on clinical use (1/22). Description of the studied population was often lacking, especially patient race/ethnicity (2/22) and mental or behavioral health (0/22). Only one study reported any item related to social determinants of health.

### Conclusion

Studies commonly do not meet rigorous methodologic standards and often fail to report pertinent details that would guide applicability. These clinical tools focus primarily on specific

**Competing interests:** The authors have declared that no competing interests exist.

disease entities and clinical variables, missing the breadth of information necessary to make a disposition determination and raise significant validation and generalizability concerns.

## Introduction

A growing area within the medical literature focuses on clinical prediction and decision tools that use variables from patient history, examination, or diagnostic tests to generate outcome-based risk stratification and/or intervention recommendations [1–3]. These tools often take the form of a simple clinical score that can be easily calculated by a bedside clinician; this is thought to provide actionable information and clinical decision support that will lead to higher quality of care through improved efficiency, greater adherence to guidelines, and standardization of care. Medical calculators that operationalize these tools in a rapid and easily accessed interface are prevalent [4, 5]. Grading criteria for these types of tools have been proposed but are not in common use [6, 7]. Thus, demonstration of appropriate use and clinical validity remains a need, particularly for the calculator tools that present recommended next therapeutic or management steps.

The decision to admit a patient to the hospital is a complicated, multifaceted phenomenon informed by contextual details related to the patient, physician, healthcare system, and overall social support availability [8–11]. In one study, nearly half of admissions from an emergency department were "strongly or moderately" influenced by one or more non-medical factors [12]. Thus, it is imperative to understand not only how clinical prediction and decision tools are employed in these types of encounters but also whether they are appropriately representative to be applied to a particular environment and situation. Here we catalogue the foundational evidence for tools that advise the decision to admit a patient to the hospital and aim to 1) identify the specific clinical scenarios and outcomes studied, 2) assess adherence to methodologic standards, and 3) determine applicability to diverse patient populations.

## Methods

### Study design

Our approach was crafted in order to reflect real-world use of risk calculators. We hypothesized that many calculator tools commonly used to inform admission or discharge decisions may not have been derived specifically for this purpose and thus would be missed by a standard literature review. Instead, we designed a pragmatic approach using a commonly available resource that describes how these tools are often used–an "end-use" search strategy as compared to an outcome- or topic- based approach.

### Medical calculator selection

We used the MDCalc online website (www.mdcalc.com) as the source for potential clinical prediction tools to review because it is a large repository of medical calculators that is free, available to all, and widely used by health professionals [13–15]. The website estimates that "as of the beginning of 2018 approximately 65% of U.S. physicians used MDcalc on a regular (weekly) basis, and millions of physicians worldwide [16]." In addition, we felt that this website provided reliable collation through its use of a peer review process for inclusion of new calculator tools and written descriptions of how each calculator is commonly used clinically that is accredited for continuing medical education.

Tools selected for inclusion in this study were those that provided recommendation for outpatient versus inpatient management, reflected in the use of any phrase such as "arrange expedited followup," "consider discharge with close followup," or "outpatient care/treatment" in its entry. We restricted to only those that included adult populations. Scoring tools for obstetrics/gynecology or pediatric populations were excluded as were tools that focused on need for intensive care admission. Both authors independently reviewed the MDCalc website in early 2021 to identify calculators appropriate for selection. These were reconciled and a final list of calculators was agreed upon as of February 15, 2021. The list of calculators included is provided in S1 Table.

## Primary studies

As described above, the MDCalc website provides not only a repository of clinically useful risk calculators but also peer reviewed contextual information for each calculator tool listed. Because clinical calculators are intended to be succinct and easy-to-use, details about where and how they were generated may not be readily apparent within the tool itself and return to the literature is necessary. Embedded within the MDCalc website display for each calculator is a section called "Evidence" which refers readers to the primary study from which the calculator was derived. The full-text primary study reference listed for each included calculator (S1 Table) was reviewed and used for the data analysis steps listed below.

## Data analysis

Data analysis in this investigation was carried out in the following steps: 1) characterize the primary studies from which the included risk calculator tools were derived, 2) evaluate the methodologic basis of the literature that underlies these risk calculator tools, and 3) assess the applicability of these risk calculator tools in broader contexts.

To address the first task of characterizing the primary studies, each primary reference study was read in full by both authors. Study methods were summarized and the country/ies in which the study was performed were captured. In addition, basic descriptors such as the clinical scenario studied, outcome measured, and study setting (e.g., Emergency Department, inpatient ward, outpatient clinic) were captured.

To address the second task of evaluating the methodologic basis of underlying literature, a standardized framework was applied. Previous analysis of and methodologic standards for clinical prediction rules have been described, first by Wasson et al in 1985 [17] and subsequently expanded by Laupacis et al in 1997 [2]. Recent literature has also championed similar approaches [18, 19]. We used the Wasson-Laupacis framework in order to systematically and rigorously describe the standards met by medical calculators included in our study. Elements were sought within the primary reference study independently by each author and then reconciled for any differences. A listing of the methodologic standards evaluated is provided below.

To address the third task of assessing applicability in broader contexts, we sought to identify what descriptive details were included in each primary reference study that would allow for other investigators to replicate the study and/or for users of the risk calculator tool to assess appropriateness for application in their clinical work. The presence of details regarding patient population (including age, sex, race/ethnicity, functional status, medical comorbidities, mental or behavioral health comorbidities, and substance use) and study setting (including location type, geographic setting, community vs. academic affiliation, size/patient volume, and rural/suburban/urban setting) was captured. Finally, because negative social determinants of health (SDOH) are associated with poor outcomes after ED discharge [20] and therefore may hold important contextual details regarding appropriateness for admission to the hospital, we

examined each primary reference for its description of any SDOH factors. While not necessarily widespread practice, there are continued calls for the integration of social care into the health care system in the literature [21] and so our approach provides a reflection of these calls to action. A position paper published by the American College of Physicians in 2018 [22] provides a listing of SDOH categories and examples in its Appendix Table which we adopted for the details of SDOH domains (economic stability, neighborhood/physical environment, education, food, community and social context, health care system) searched for in each primary study reference. Beyond identifying whether details of patient population, study setting, or SDOH were described in the primary reference study, we also identified if they were incorporated into the corresponding risk calculator tool.

**Methodologic standards evaluated.** Each primary reference study was analyzed with respect to methodologic standards in 10 domains: outcome (definition, clinical importance, and blind assessment), predictive variables (identification and definition, blind assessment), important patient characteristics described, study site described, mathematical techniques described, results of the rule described, reproducibility (of predictive variables, of the rule), sensibility (clinically sensible, easy to use, probability of disease described, course of action described), prospective validation, and effects of clinical use prospectively measured. Definitions of these domains have been previously published [2].

## Methodologic standards requiring interpretation

In some instances, there were methodologic standards that required some interpretation on the part of the authors to determine whether present or absent. We determined a priori definitions for meeting these standards. For example, we classified that the "important patient characteristics described" methodology standard would be met if the primary reference study included any patient information beyond age, sex, or medical comorbidities (as we posited that hospital admission requires a more comprehensive, holistic view of the patient's health and context). Likewise, in order to meet the requirement for "study site described" we required inclusion of any specifics beyond location type (ED, clinic, hospital) and geographic setting (country and/or region). Details of which specific items were included in patient characteristics and study settings was then incorporated into the applicability assessment.

## Statistical analysis

There were a total of 22 calculator tools selected for inclusion in this study. As described above, the primary reference study for each calculator tool served as the primary source for data analysis. Descriptive statistics are provided by count (how many met the criterion or standard of interest) and percentage (of the total 22).

## Results

A total of 22 calculators provided hospital admission recommendation for the following discrete clinical presentations: chest pain/suspected ACS (7), pulmonary embolism (4), community-acquired pneumonia (2), heart failure (2), GI bleed (2), febrile neutropenia (2), syncope (1), TIA (1), and suspected appendicitis (1). A summary of outcomes measured and study setting for each clinical scenario is provided in Table 1. Reporting of methodologic standards is provided in Table 2 and described in detail below.

## Outcomes and predictive variables

Each study reported clinically important outcomes. Outcomes were typically a measure of potential patient risk either specifically defined as patient mortality (6/22) or an aggregate

**Table 1. Clinical scenario, outcome measured, and study setting for clinical risk scoring tools.**

| Clinical Scenario (n) | Outcome Measured (n) | Study Setting (n) |
|---|---|---|
| Chest Pain/Suspected ACS (7) | Serious Outcome* (6), CAD (1) | ED (6), Primary Care (1) |
| Pulmonary Embolism (4) | Mortality (3), Serious Outcome (1) | ED (2), Inpatient (2) |
| Heart Failure (2) | Mortality (1), Serious Outcome (1) | ED (2) |
| Community Acquired Pneumonia (2) | Mortality (2) | Inpatient (2) |
| Febrile Neutropenia (2) | Serious Outcome (2) | Inpatient (2) |
| GI Bleed (2) | Serious Outcome (2) | Inpatient (2) |
| Syncope (1) | Serious Outcome (1) | ED (1) |
| Suspected Appendicitis (1) | Confirmed Appendicitis (1) | Inpatient (1) |
| Transient Ischemic Attack (1) | Serious Outcome (1) | Population-Based, including ED and Clinic (1) |

Each row summarizes outcomes measured and study setting for the corresponding clinical scenario

ACS = Acute Coronary Syndrome; CAD = Coronary Artery Disease; ED = Emergency Department

*Most Serious Outcomes In Chest Pain Category Used Major Adverse Cardiac Event (MACE)

assessment of serious outcome such as adverse event/complication which could include mortality (14/22). A small number of studies (2/22) used confirmatory diagnosis (e.g., confirmed appendicitis for patients with clinically suspected appendicitis) as the outcome of interest. Outcomes were sufficiently defined in all but one study which did not provide acute myocardial

**Table 2. Methodological standards applied to calculator tool primary reference (total: 22 studies).**

| Methodologic Standard | Reports that Met Standard, n/22 (%) |
|---|---|
| Outcome | |
| Clinical importance | 22/22 (100%) |
| Definition | 21/22 (95%) |
| Blind assessment | 8/22 (36%) |
| Predictive variables | |
| Identification and definition | 17/22 (77%) |
| Blind assessment | 16/22 (73%) |
| Important patient characteristics described | 12/22 (55%) |
| Study site described | 14/22 (64%) |
| Mathematical techniques described | 22/22 (100%) |
| Results of the rule described | 20/22 (91%) |
| Reproducibility | |
| Of predictive variables | 5/22 (23%) |
| Of the rule | 1/22 (5%) |
| Prospective validation | 12/22 (55%) |
| Sensibility | |
| Clinically sensible | 22/22 (100%) |
| Easy to use | 17/22 (77%) |
| Probability of disease described | 22/22 (100%) |
| Course of action described | 22/22 (100%) |
| Effects of clinical use prospectively measured | 1/22 (5%) |

infarction diagnostic criteria. Only a small proportion of studies (8/22) conveyed blind assessment of outcomes as part of the study protocol. Most studies (17/22) sufficiently identified and defined the predictive variables used in their scoring models and most (16/22) described using blinded assessment of the predictive variables.

## Patient populations, study sites, and social determinants of health

Study populations were most commonly characterized by patient age (21/22) and sex (20/22), though one study did not include any such information. Preexisting medical comorbidities were also commonly referenced (21/22). Additional patient characteristics beyond age, sex, and medical comorbidities were reported in around half of the studies (13/22) though there were significant limitations to what was included. When substance use was described (7/22), it only referenced cigarette smoking and not drug or alcohol use. Functional status was not commonly included (6/22) and was characterized by the Eastern Cooperative Oncology Group (ECOG) score in two studies, nursing home residency in two studies, and immobility or paralysis in two studies. Patient race or ethnicity was rarely included (2/22) and preexisting mental or behavioral health was never described.

All studies indicated the location of investigation (ED, inpatient, clinic) and all but two specified the geographic area (country or region) in which the study was conducted. Just over half of the studies (14/22) met criterion of "study site described" by inclusion of another detail beyond these: eleven studies included description of community or academic setting; six included description of size or volume of patients seen; and five specified whether sites were located in urban, suburban, or rural settings.

Only one study incorporated any item related to social determinants of health. There were no studies that described items in the categories of economic stability, neighborhood/physical environment, education, food, or health care systems. The study that referenced community and social context did so via its exclusion criteria of individuals with a "medical or social reason for treatment in the hospital for more than 24 hours (infection, malignancy, no support system) [23]." Of note, 10% of the patients screened for this study were excluded based on "social needs."

Fig 1 depicts the study details presented above regarding patient, setting, and SDOH descriptions in addition to inclusion of these features within the final calculator scoring tool itself. For patient population details, medical comorbidities (17/22) were most commonly included in the score followed by age (14/22), sex (6/22), functional status (2/22), and

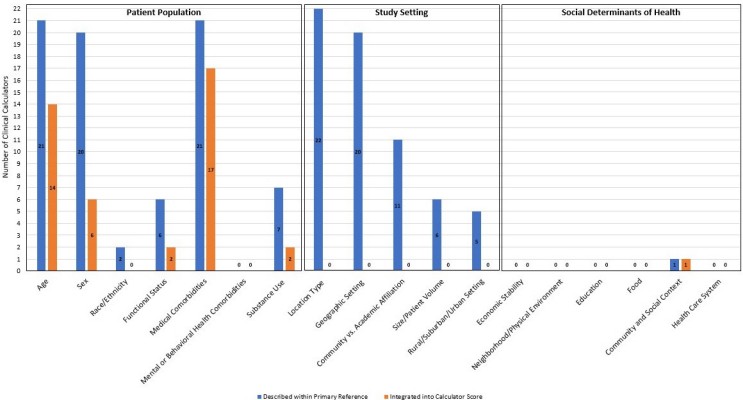

**Fig 1. Primary reference and calculator score inclusion of patient population, study setting, and social determinants of health details.**

substance use (2/22). Patient race/ethnicity and mental or behavioral health comorbidities were not included. No study setting details were included in the scores. Only one calculator included any SDOH in the score.

## Mathematical techniques and results of the rule

All reports of statistical techniques used were reasonably described. Most studies (18/22) used some form of multivariate analysis; 16 used logistic regression and two used recursive partitioning. Of the rest, two studies reported only univariate analysis, one study tested a predefined accelerated diagnostic protocol and reported sensitivity/specificity and negative/positive predictive values for protocol components both individually and collectively, and one study reported the rate of adverse events measured prospectively when a protocol explicitly directing outpatient management was implemented clinically. At the individual predictor level, correlation with outcome was reported by odds ratio in ten studies and by beta coefficients in one study. Four studies involved a higher ratio of cases to covariates than the recommended 1:10. In two of the 22 studies it was not possible to tell how many total predictor variables were considered for inclusion.

In total, 20/22 studies described the results of the rule in some way. Sensitivity, specificity, and/or predictive values were explicitly reported in 15 studies. Receiver operating characteristics and/or c-statistics were used in 15 studies as part of assessment of diagnostic accuracy, comparison between derivation and validation cohorts, and/or comparison with other existing risk scoring models.

## Reproducibility and prospective validation

Reproducibility was not commonly assessed in these studies. A small number (5/22) reported a process for verifying reproducibility of individual predictive variables by different data abstractors. Only one study reported the reproducibility of achieving the same final result between different users of the rule. Prospective validation was heterogeneously performed. Just over half (12/22) of the studies reported prospective validation using a different population than the derivation cohort; this included three studies that enrolled similar populations but at a different time period. Two studies described retrospective validation using a different study population than derivation. Four studies split the initial group of participants into derivation and validation subsets. Four studies performed no validation.

## Sensibility and effects of clinical use

Overall, the clinical tools generated appeared to be clinically sensible. All studies corresponded with a readily-available website tool (since this was criteria for entry into the study), however 5/22 studies included 10 or more elements in the final scoring tool, which we did not consider easy to use. Of note, some of the studies included were attempts at simplification of these existing more complicated scoring tools (e.g., sPESI and PESI; CURB-65 and PSI/PORT).

## Course of action described and effects of clinical use

Each study included some form of recommendation related to inpatient versus outpatient management though how integrated this was into the output of the risk calculator tool varied. For 12/22 studies, the outcome of the rule itself provided this guidance (e.g., at a particular score threshold, outpatient management is recommended). For the remainder of the studies outpatient vs. inpatient guidance was suggested as a means of using the result of the rule clinically. One study provided prospective measurement of score impact on clinical decision-making via

inclusion/exclusion rates when implemented in a clinical setting; six studies reported a hypothetical estimated effect if their respective scoring tools were to be implemented.

## Discussion

As clinical risk tools become increasingly common, ensuring their quality as well as appropriate application of their results is of paramount importance, particularly as there is continued interest in embedding clinical scores within the electronic medical record, often in an automated fashion and integrated into clinical decision support mechanisms [24, 25]. We found that calculator tools used to inform hospital admission decisions rarely are built upon evidence studying the intervention of hospitalization itself and that there is wide variation in study design and settings. In addition, we found that research methodologic standards are inconsistently applied in this body of literature and that there are gaps in reporting of study details that make evaluation of applicability to diverse patient populations challenging.

In their 2019 perspective on the proliferation of clinical risk tools, Challener, Prokop, and Abu-Saleh argued "Clinical scoring systems should be evaluated on quality and clinical benefit. The quality of a score depends on the method of its development, the rigor of subsequent validation, and its performance characteristics [7]." Several concerns in these domains arise from our evaluation. First, the presumed benefit of hospitalization in the recommendations was often based on extrapolation from the risk of adverse events or mortality. This practice proves problematic, as described by Schenkel and Wyer: "It is not evident that because a patient may die that hospitalization will reduce that likelihood, nor is it evident that a patient likely to live will not benefit from hospital care [26]." Furthermore, assessing an outcome while the patient is already admitted as several of the studies did complicates the findings because every participant received the benefit of hospitalization.

In addition, comparing outcomes and making disposition recommendations based on the severity of disease process alone—as the risk tools in this report do—removes the impact of the patient's context and surrounding environment from the risk/benefit assessment. A variety of factors have been associated with increased risk for hospitalization including social, cognitive, and functional deficits [27], even with a low-risk clinical score [28]. Thus, these risk calculators may overemphasize clinical variables while avoiding integration of the types of complex factors that have been shown to drive admission practices [7, 9, 11, 29]. While it can be argued that these tools should remain purely clinically oriented and that physicians and other practitioners are called upon to identify if a tool is applicable to a specific case, the lack of information reported in these studies such as details of patient population, study setting, or social determinants of health makes this challenging in practice.

Finally, there are significant questions about the real-world utility and potential unintended consequences of these tools as well as if it is even possible to appropriately assess their outcomes. The initial derivation of a risk score understandably may not report prospective validation of the rule or measurement of its effect when implemented in clinical care and additional prospective validation studies are often needed. With tools related to hospitalization decisions, however, this is potentially fraught and should be carefully considered. The landscape of inpatient versus outpatient care is changing with increased in-home care and/or remote monitoring as well as targeted relationships between hospitals and post-acute care facilities to provide alternatives to hospitalization [30–32]. Conditions which in previous time periods would have been appropriately managed by hospitalization may now have viable alternative locations of care. Therefore, any hospitalization scoring tool is inherently limited in its generalizability and should only be applied to similar contextual environments. Quite simply, there are significant

validation issues for these calculator tools and any model built to predict need for hospitalization is not likely to provide appropriate guidance for clinicians at-large.

This study uses the MDCalc website as a convenience sample and thus is not necessarily reflective of the entirety of clinical risk calculators guiding hospitalization decision-making. Also, by specifically evaluating the primary reference for each calculator tool as listed on the website, the findings are subject to any inherent bias in MDCalc's selection and review process. We chose this approach for consistency across calculator tools and because it pragmatically reflects how many users of the tools access them and are directed to the literature. We did not directly assess clinician behavior in terms of how they identify need for a calculator, access the calculator tool itself, assess the applicability of a calculator to a specific case, and integrate its output into their overall decision-making. These would be important areas of future exploration.

Strengths of this study include use of a novel, "end-use" search strategy in which we used a functional, pragmatic approach to search for how clinical risk calculator tools are being used in practice, rather than what they may have been intended to measure. This allows for capture of tools that may otherwise be missed when searching by outcome or clinical disease entity. This is an approach that could be considered for other end-use assessments.

## Conclusion

When examining the literature underlying clinical risk scoring tools that advise admission or discharge decision-making, we found that methodologic standards are not universally met and information to guide applicability is lacking. These tools focus primarily on specific disease entities and clinical variables which may not encompass the breadth of information necessary to make a disposition determination. Taken together, our results do not support broad use of these calculators for the purpose of determining need for hospitalization.

## Supporting information

**S1 Table. Key characteristics of clinical risk calculator tools and primary references.**
(DOCX)

## Author Contributions

**Conceptualization:** Neeloofar Soleimanpour, Maralyssa Bann.

**Data curation:** Neeloofar Soleimanpour, Maralyssa Bann.

**Formal analysis:** Neeloofar Soleimanpour, Maralyssa Bann.

**Investigation:** Maralyssa Bann.

**Methodology:** Maralyssa Bann.

**Project administration:** Maralyssa Bann.

**Supervision:** Maralyssa Bann.

**Writing – original draft:** Neeloofar Soleimanpour, Maralyssa Bann.

**Writing – review & editing:** Neeloofar Soleimanpour, Maralyssa Bann.

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
