## [Decision Letter · Decision Letter 0]

10 Oct 2022

PONE-D-22-16680Clinical Risk Calculators Informing the Decision to Admit: A Methodologic Evaluation and Assessment of ApplicabilityPLOS ONE

Dear Dr.Maralyssa Bann,

Thank you for submitting your manuscript to PLOS ONE. After careful consideration, we feel that it has merit but does not fully meet PLOS ONE’s publication criteria as it currently stands. Therefore, we invite you to submit a revised version of the manuscript that addresses the points raised during the review process.

ACADEMIC EDITOR:

The study is interesting however major revisions are needed to be able to accept it for publication.

In particular:

1. A careful review of the methodological analysis is needed and to rewrite the methods section adequately

2. Careful review of the statistical analysis

3. Summarize the results by inserting graphs to make them clearer and more intelligible

4. In the discussion highlight how your results are generalizable and usable in other contexts.

5. Please respond to the reviewers' requests point by point.

The decision is justified on PLOS ONE’s publication criteria.

We look forward to receiving your revised manuscript.

Kind regards,

Filomena Pietrantonio

Academic Editor

PLOS ONE

Journal Requirements

Additional Editor Comments:

The study is interesting however major revisions are needed to be able to accept it for publication.

In particular:

1. A careful review of the methodological analysis is needed and to rewrite the methods section adequately

2. Careful review of the statistical analysis

3. Summarize the results by inserting graphs to make them clearer and more intelligible

4. In the discussion highlight how your results are generalizable and usable in other contexts.

5. Please respond to the reviewers' requests point by point.

Reviewers' comments:

Reviewer's Responses to Questions

**Comments to the Author**

1. Is the manuscript technically sound, and do the data support the conclusions?

Reviewer #1: Partly

Reviewer #2: Yes

2. Has the statistical analysis been performed appropriately and rigorously? 

Reviewer #1: No

Reviewer #2: N/A

3. Have the authors made all data underlying the findings in their manuscript fully available?

Reviewer #1: No

Reviewer #2: Yes

4. Is the manuscript presented in an intelligible fashion and written in standard English?

Reviewer #1: Yes

Reviewer #2: Yes

5. Review Comments to the Author

Reviewer #1: Dear Authors, thank you for your work. Unfortunately, methodological analysis is puzzling, and results are not intelligibles. Please, rewrite "Methods" sections. Use a medical statistical software for statistics analysis

Reviewer #2: the paper is interesting and well written. I have some minor concerns: it would be useful to sinthesize your results in graphic format. for example you could cluster the dimanesions for evaluation in two or three groups and then build up a catterplot so that each risk indicator is placed within the cartesian space of the graph. In this way your results would be more intellegible and interpretable.

6. PLOS authors have the option to publish the peer review history of their article (what does this mean?). If published, this will include your full peer review and any attached files.

Reviewer #1: No

Reviewer #2: No

---

## [Author Response · Author response to Decision Letter 0]

23 Nov 2022

Author’s Response to Decision Letter for Manuscript PONE-D-22-16680 entitled “Clinical Risk Calculators Informing the Decision to Admit: A Methodologic Evaluation and Assessment of Applicability”

Dear Editor and Reviewers,

We appreciate the ability to revise this manuscript and have reviewed and addressed reviewer comments below. We welcome any additional feedback.

Best regards,

Neeloofar Soleimanpour and Maralyssa Bann

Author’s Reply to Editor Comments

1. A careful review of the methodological analysis is needed and to rewrite the methods section adequately

We recognize that our study design and data analysis may have been confusing and appreciate the opportunity to clarify this further. The Methods section has been substantially revised. 

To begin, we have more explicitly described the source of and importance of the primary studies in its own labeled portion of the Methods section: 

Primary Studies

As described above, the MDCalc website provides not only a repository of clinically useful risk calculators but also peer reviewed contextual information for each calculator tool listed. Because clinical calculators are intended to be succinct and easy-to-use, details about where and how they were generated may not be readily apparent within the tool itself and return to the literature is necessary. Embedded within the MDCalc website display for each calculator is a section called “Evidence” which refers readers to the primary study from which the calculator was derived. The full-text primary study reference listed for each included calculator (eTable 1) was reviewed and used for the data analysis steps listed below.

We then much more specifically describe the 3 steps of our Methods under Data Analysis:

Data Analysis

Data analysis in this investigation was carried out in the following steps: 1) characterize the primary studies from which the included risk calculator tools were derived, 2) evaluate the methodologic basis of the literature that underlies these risk calculator tools, and 3) assess the applicability of these risk calculator tools in broader contexts. 

To address the first task of characterizing the primary studies, each primary reference study was read in full by both authors. Study methods were summarized and the country/ies in which the study was performed were captured. In addition, basic descriptors such as the clinical scenario studied, outcome measured, and study setting (e.g., Emergency Department, inpatient ward, outpatient clinic) were captured.

To address the second task of evaluating the methodologic basis of underlying literature, a standardized framework was applied. Previous analysis of and methodologic standards for clinical prediction rules have been described, first by Wasson et al in 198517 and subsequently expanded by Laupacis et al in 1997.2 Recent literature has also championed similar approaches.18,19 We used the Wasson-Laupacis framework in order to systematically and rigorously describe the standards met by medical calculators included in our study. Elements were sought within the primary reference study independently by each author and then reconciled for any differences. A listing of the methodologic standards evaluated is provided below.

To address the third task of assessing applicability in broader contexts, we sought to identify what descriptive details were included in each primary reference study that would allow for other investigators to replicate the study and/or for users of the risk calculator tool to assess appropriateness for application in their clinical work. The presence of details regarding patient population (including age, sex, race/ethnicity, functional status, medical comorbidities, mental or behavioral health comorbidities, and substance use) and study setting (including location type, geographic setting, community vs. academic affiliation, size/patient volume, and rural/suburban/urban setting) was captured. Finally, because negative social determinants of health (SDOH) are associated with poor outcomes after ED discharge20 and therefore may hold important contextual details regarding appropriateness for admission to the hospital, we examined each primary reference for its description of any SDOH factors. While not necessarily widespread practice, there are continued calls for the integration of social care into the health care system in the literature21 and so our approach provides a reflection of these calls to action. A position paper published by the American College of Physicians in 201822 provides a listing of SDOH categories and examples in its Appendix Table which we adopted for the details of SDOH domains (economic stability, neighborhood/physical environment, education, food, community and social context, health care system) searched for in each primary study reference. Beyond identifying whether details of patient population, study setting, or SDOH were described in the primary reference study, we also identified if they were incorporated into the corresponding risk calculator tool.

We also specify how we defined certain methodologic standards a priori:

Methodologic Standards Requiring Interpretation

In some instances, there were methodologic standards that required some interpretation on the part of the authors to determine whether present or absent. We determined a priori definitions for meeting these standards. For example, we classified that the “important patient characteristics described” methodology standard would be met if the primary reference study included any patient information beyond age, sex, or medical comorbidities (as we posited that hospital admission requires a more comprehensive, holistic view of the patient’s health and context). Likewise, in order to meet the requirement for “study site described” we required inclusion of any specifics beyond location type (ED, clinic, hospital) and geographic setting (country and/or region). Details of which specific items were included in patient characteristics and study settings was then incorporated into the applicability assessment. 

2. Careful review of the statistical analysis

Thank you for this suggestion. We have now explained our descriptive statistics in more detail (count and percentage of the total number of studies) with the following text:

Statistical Analysis

There were a total of 22 calculator tools selected for inclusion in this study. As described above, the primary reference study for each calculator tool served as the primary source for data analysis. Descriptive statistics are provided by count (how many met the criterion or standard of interest) and percentage (of the total 22).

3. Summarize the results by inserting graphs to make them clearer and more intelligible

Thank you for this suggestion. We have reviewed the data contained in the current tables and have converted Table 3 into a graph. See Figure 1.

4. In the discussion highlight how your results are generalizable and usable in other contexts.

We would like to be careful to note that the findings of this investigation reveal that there is not enough methodologic rigor to support the broad use of risk calculator tools for admission decision-making. The question of generalizability pertains more to the methods we employed here so the Discussion section now includes the following text:

Strengths of this study include use of a novel, “end-use” search strategy in which we used a functional, pragmatic approach to search for how clinical risk calculator tools are being used in practice, rather than what they may have been intended to measure. This allows for capture of tools that may otherwise be missed when searching by outcome or clinical disease entity. This is an approach that could be considered for other end-use assessments.

5. Please respond to the reviewers' requests point by point.

See below.

Reviewer #1: 

Dear Authors, thank you for your work. Unfortunately, methodological analysis is puzzling, and results are not intelligibles. Please, rewrite "Methods" sections. Use a medical statistical software for statistics analysis

Thank you for your comments, We hope that the substantial revision of the Methods section as described above may help with understanding the analysis and its results. 

Use a medical statistical software for statistics analysis

Thank you for the opportunity to clarify this point. Because the statistics presented are counts and percentages, they are able to be calculated without more in-depth software. More intricate analysis using correlational or hypothesis-testing statistical inferences does not fit this data and is beyond the scope of the current investigation. We hope that the clarification of methods and statistical analysis will help this clarification. 

Reviewer #2: the paper is interesting and well written. I have some minor concerns: it would be useful to sinthesize your results in graphic format. for example you could cluster the dimanesions for evaluation in two or three groups and then build up a catterplot so that each risk indicator is placed within the cartesian space of the graph. In this way your results would be more intellegible and interpretable.

Thank you for this suggestion. We agree that visual representation can add significantly to the ability to understand findings. In this case, we attempted to create a scatterplot as suggested but were unable to capture all of the data currently in table form in this manner. Some of the meaning was lost. We have added the following line underneath Table 1 in order to provide additional description:

Each row summarizes outcomes measured and study setting for the corresponding clinical scenario

In addition, we have converted what was previously Table 3 into graph form in order to highlight the stark differences between the domains as well as what was captured in primary reference versus what was included in the calculator tool itself. We feel that this graphical representation adds useful nuance and makes the point easier to understand. We appreciate the reviewer’s suggestion.

---

## [Decision Letter · Decision Letter 1]

5 Dec 2022

Clinical Risk Calculators Informing the Decision to Admit: A Methodologic Evaluation and Assessment of Applicability

PONE-D-22-16680R1

Dear Dr. Maralyssa Bann, 

We’re pleased to inform you that your manuscript has been judged scientifically suitable for publication and will be formally accepted for publication once it meets all outstanding technical requirements.

Kind regards,

Filomena Pietrantonio

Academic Editor

PLOS ONE

Additional Editor Comments (optional):

The authors have addressed all comments and the manuscript is now suitable for publication

Reviewers' comments:

Reviewer's Responses to Questions

**Comments to the Author**

1. If the authors have adequately addressed your comments raised in a previous round of review and you feel that this manuscript is now acceptable for publication, you may indicate that here to bypass the “Comments to the Author” section, enter your conflict of interest statement in the “Confidential to Editor” section, and submit your "Accept" recommendation.

Reviewer #1: All comments have been addressed

Reviewer #2: All comments have been addressed

2. Is the manuscript technically sound, and do the data support the conclusions?

Reviewer #1: Yes

Reviewer #2: Yes

3. Has the statistical analysis been performed appropriately and rigorously? 

Reviewer #1: Yes

Reviewer #2: Yes

4. Have the authors made all data underlying the findings in their manuscript fully available?

Reviewer #1: Yes

Reviewer #2: No

5. Is the manuscript presented in an intelligible fashion and written in standard English?

Reviewer #1: Yes

Reviewer #2: Yes

6. Review Comments to the Author

Reviewer #1: dear authors, thank you for your work and your revisions. now your work is finally read to publish!!

Reviewer #2: the authors have addressed all comments and the manuscript is now suitable for publication on plos one

7. PLOS authors have the option to publish the peer review history of their article (what does this mean?). If published, this will include your full peer review and any attached files.

Reviewer #1: No

Reviewer #2: No

---

## [Editor Report · Acceptance letter]

12 Dec 2022

PONE-D-22-16680R1 

Clinical Risk Calculators Informing the Decision to Admit: A Methodologic Evaluation and Assessment of Applicability 

Dear Dr. Bann:

I'm pleased to inform you that your manuscript has been deemed suitable for publication in PLOS ONE. Congratulations! Your manuscript is now with our production department. 

Kind regards, 

on behalf of

Dr. Filomena Pietrantonio 

Academic Editor

PLOS ONE